

# Technical note: On seasonal variability of the M$_2$ tide

Richard D. Ray[1]

[1]Geodesy & Geophysics Lab., NASA Goddard Space Flight Center, Greenbelt, Maryland, USA

**Correspondence:** Richard Ray (richard.ray@nasa.gov)

**Abstract.** Seasonal variability of the M$_2$ ocean tide can be detected at many ports, perhaps most. Examination of the cluster of tidal constituents residing within the M$_2$ tidal group can shed light on the physical mechanisms underlying seasonality. In broadest terms these are astronomical, frictional/advective interactions, and climate processes; some induce annual modulations, some semiannual, in amplitude, phase, or both. This note reviews how this occurs and gives an example from each broad category. Phase conventions and their relationship to causal mechanisms, as well as nomenclature, are also addressed.

## 1 Introduction

It has long been noticed (Darwin, 1907; Corkan, 1934) that ocean tide constituents at some ports may experience significant seasonal variations. Especially noteworthy are large modulations discovered in some polar regions (e.g., Godin, 1986; Rotermund et al., 2021; Bij de Vaate et al., 2021). Significant modulations can also occur in lower latitudes, both regionally (Kang et al., 2002) and especially locally (e.g., Foreman et al., 1995). In fact, nearly all coastal tides show at least a small seasonal modulation (Pugh and Woodworth, 2014).

Climate-driven processes capable of inducing seasonal changes in barotropic tides are myriad: variability in ocean stratification (Müller, 2012), variability in ice cover (Prinsenberg, 1988), seasonal runoff and changes in river discharge (Guo et al., 2015), tide-surge interactions from predominantly wintertime storms (Pugh and Woodworth, 2014). Similar processes and others can induce seasonal perturbations in baroclinic tides detectable in surface measurements (Ray and Zaron, 2011; Zhao, 2021). The extent that the astronomical tidal potential plays in seasonality—usually small, but potentially important where tide amplitudes are large—is often overlooked. Some confusion on the issue has been recently clarified by Du and Yu (2021).

To help unravel observations it is useful to revisit the spectral characteristics behind seasonal variability. Understanding how certain spectral lines in observed sea level arise often points to possible physical mechanisms at work. The purpose of this note is to review this topic, focusing solely on the principal semidiurnal tide M$_2$. Much of what follows is hardly new; the purpose is to review and clarify, including even the nomenclature used. Whether seasonal variability is annual or semiannual, and whether it occurs in amplitude or phase or both, are obviously important aspects of variability; seeing one type of modulation rather than another can narrow the list of causes.





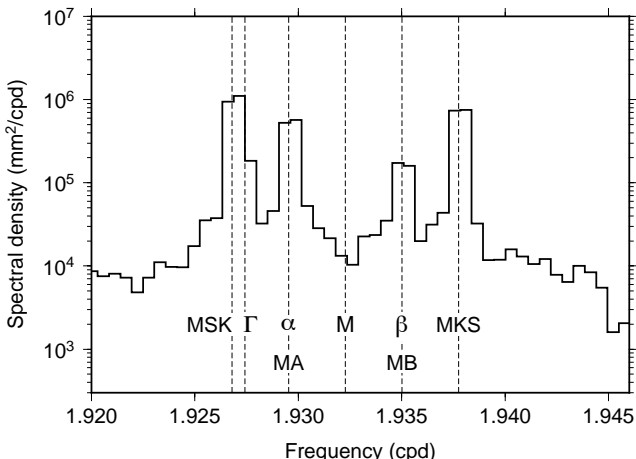

**Figure 1.** Spectrum of sea level at Saint-Malo, on the north coast of France, focusing on the $M_2$ group, but with the central $M_2$ constituent estimated and removed to better delineate the much smaller sidelines. The spectrum is based on 16 years of data. After spectral smoothing, the frequency resolution is approximately 0.2 cpy (or 0.0005 cpd), insufficient to clearly separate $MSK_2$ from $\Gamma_2$. The seasonal modulation of $M_2$ at Saint-Malo is evidently dominated by the two frictional compound tides, although $\alpha_2$ is also important—see discussion in Section 3.2.

## 2  The $M_2$ tidal group

Recall the technical definitions (e.g., Munk and Cartwright, 1966) of 'tidal group'—a cluster of spectral lines with the same first two Doodson numbers—and 'tidal constituent'—a cluster with the same first three Doodson numbers. Tidal groups are separated in frequency by about 1 cycle per month, constituents by about 1 cycle per year. So when one speaks of seasonal variability of the $M_2$ tide, one must be speaking, in a sense, of the $M_2$ tidal group. It is the variability seen, for example, in a series of monthly estimates of $M_2$.

The main constituents within the $M_2$ group are listed in Table 1 in order of frequency. Included are tides generated by the astronomical potential (Cartwright and Edden, 1973; Hartmann and Wenzel, 1995), compound tides generated by shallow-water processes, and two annual sidelines ($MA_2$ and $MB_2$) commonly employed to effect an annual modulation from the wide range of possible climate processes. Climate processes are broadband, but they do commonly display a large spectral peak at once per year, and the two sidelines attempt to account for that. An example of these spectral lines in a real sea-level spectrum

is shown in Fig. 1, computed from the tide-gauge measurements at Saint-Malo, France (and used further below).

Since all tides in the table are lunar, they all have 18.6-y nodal sidelines. These will be ignored in the present context, but they are obviously important for tidal prediction.

The constituent arguments given in the table are in the form of the standard mean longitudes that Doodson (1921) found so useful for ordering the whole tidal catalog. Simple expressions, linear in time for present-day tides, are readily available to

evaluate the longitudes, and thus the tidal arguments, for any particular time (e.g., Pugh and Woodworth, 2014, p. 68).





**Table 1.** Tidal constituents within the $M_2$ group.

| Constituent | Source | Argument | Freq. (deg/h) | Relative amp. |
|---|---|---|---|---|
| $MSK_2$ | Friction | $2\tau - 2h$ | 28.901967 | – |
| $\Gamma_2$ | Gravitation | $2\tau - 2h + 2p + \pi$ | 28.911251 | 0.00301 |
| $MA_2$ | Climate | $2\tau - h$ | 28.943036 | – |
| $\alpha_2$ | Gravitation | $2\tau - h + p' + \pi$ | 28.943038 | 0.00345 |
| $M_2$ | Gravitation | $2\tau$ | 28.984104 | 1.00000 |
| $\beta_2$ | Gravitation | $2\tau + h - p'$ | 29.025171 | 0.00304 |
| $MB_2$ | Climate | $2\tau + h$ | 29.025173 | – |
| $\delta_2$ | Gravitation | $2\tau + 2h$ | 29.066242 | 0.00114 |
| $MKS_2$ | Friction | $2\tau + 2h$ | 29.066242 | – |

$\tau$ – mean lunar time

$h$ – mean longitude of sun

$p$ – mean longitude of lunar perigee

$p'$ – mean longitude of solar perigee, currently about $283°$

All the tides of Table 1 differ in frequency from the central $M_2$ tide by either one or two cycles per year (cpy). They thus act to modulate the amplitude and/or phase of $M_2$ at those frequencies—the source, at least in spectral terms, of the seasonality of $M_2$. Most of the tidal arguments differ from $M_2$ by either $\pm h$ or $\pm 2h$, so the modulation is tied to the sun's declination, zero at the equinox; the speed of $h$ is one cycle per tropical year. The two astronomical tides, $\alpha_2$ and $\beta_2$, differ from $M_2$ by $\pm \ell'$ where 45 $\ell' = h - p'$ is the sun's mean anomaly, zero at perihelion, so the modulation is tied to the sun's distance; the speed of $\ell'$ is one cycle per anomalistic year. The astronomical tide $\Gamma_2$ also depends on the position of the moon's perigee $p$. In an average sense, over many years, the contribution from $\Gamma_2$ to $M_2$ seasonality mostly cancels, because $p$ (period 8.85 y) varies from one year to the next.

The frequencies of Table 1 are given with sufficient precision to show the tiny differences stemming from tropical versus 50 anomalistic years, which are almost identical because the slowly moving $p'$ requires 209 centuries to complete one revolution. So the annual pairs $MA_2$, $MB_2$ and $\alpha_2, \beta_2$ take practically the same frequencies, and either pair may be used in a tidal analysis. But because their phases differ, the pairs cannot be interchanged; analysis and any subsequent prediction must maintain consistency.

Each class of constituent is now addressed in more detail.

## 2.1 Astronomical tides

The four gravitational sidelines are all very small, only about 0.3% of the primary, or smaller. With arguments depending on the mean longitude of the sun, they evidently arise from the sun's third-body perturbations of the lunar orbit. In particular, $\alpha_2, \beta_2$ arise from what in lunar theory is called the "annual equation," which refers to an expansion or compaction of the moon's





orbit depending on whether the sun is at perihelion or aphelion, respectively. With a change in the orbital radius there is a

corresponding change in the moon's angular velocity and thus longitude. The variation in longitude is given approximately by

$-669'' \sin \ell'$ (Brouwer and Clemence, 1961, p. 329). The radial variation is $49 \cos \ell'$ km.

Because the ocean cannot support extremely high-$Q$ resonances, the ocean's response (admittance) to gravitational forcing must be nearly constant across the small frequency band of the $M_2$ group. In that case, the gravitational components of the four sidelines can all be inferred from measurements of the central $M_2$ primary. Their amplitudes must be in the same proportion

as the relative amplitudes of Table 1 and their phases must be nearly identical to the phase of $M_2$. It may be possible that this rule is violated in locations where large $M_2$ currents act through nonlinear dissipation to suppress the sidelines; this has been observed to happen for the $M_2$ nodal sideline (Ku et al., 1985).

Aside from this one exception, the induced seasonal modulations in $M_2$ from the astronomical sidelines are easily worked out. At a location with mean $M_2$ amplitude $A$ and phase lag $G$, the combined elevation of the group is

$$
\begin{aligned}
\zeta_M(t) \quad = \quad & A\{\cos(2\tau - G) \\
& + c_\alpha \cos(2\tau - \ell' + \pi - G) \\
& + c_\beta \cos(2\tau + \ell' - G) \\
& + c_\delta \cos(2\tau + 2h - G)\}
\end{aligned}
$$

where $c_i$ are the relative amplitudes from Table 1, and the $\Gamma_2$ constituent has been dropped for reasons noted above. In analogy

with the usual approach for handling nodal modulations, this expression may be written as a single modulated wave:

$$
\zeta_M(t) = A f(t) \cos[2\tau - G + u(t)] \tag{1}
$$

where functions $f, u$ vary throughout the year as periodic functions of $h$ (or $\ell'$). Expanding the trigonometric functions and gathering like terms in $\cos G$ and $\sin G$ lead to

$$
\begin{aligned}
f \cos u \quad = \quad & 1 - (c_\alpha - c_\beta) \cos \ell' + c_\delta \cos 2h \tag{2}
\end{aligned}
$$

$$
f \sin u \quad = \quad (c_\alpha + c_\beta) \sin \ell' + c_\delta \sin 2h
$$

or

$$
f \approx 1 - 0.00041 \cos \ell' + 0.00114 \cos 2h \tag{3}
$$

for the amplitude modulation, and

$$
u \approx 0.372° \sin \ell' + 0.065° \sin 2h \tag{4}
$$

for the phase modulation. The amplitude modulation is insignificant; because the main coefficients in (2) cancel, the second term from $\delta_2$ is actually larger than the first term, but still very small. For the phase modulation in (4), the first term peaks at $\ell' = 90°$, which is early April, and the smaller second term shifts that peak to mid-April. So the observed phase lag of $M_2$ ($u$




subtracts from $G$) takes its minimum value in mid-April and its maximum in mid-September. It is interesting to note that the first term in (4) is exactly twice the moon's longitude modulation of $669'' \sin \ell'$, alluded to above as arising from the annual equation in the sun's perturbation of the lunar orbit (it is twice, because $M_2$ is semidiurnal).

At most tide gauges, the gravitational components of any seasonal variability in the observed $M_2$ will thus comprise only a minor part. Observed variability will likely differ significantly from what might be inferred from the $M_2$ admittance or as modeled by Eqs. (3–4). Nonetheless, when only the astronomical modulation is important and the $M_2$ amplitude is moderately large, a phase modulation of $0.7°$ is easily seen; an example is given in Section 3.1.

## 2.2 Compound tides

Aside from the two compound tides listed in Table 1, there are possibly others that fall within the $M_2$ group (Simon, 2013). The most important is $OP_2$, exactly coinciding with $MSK_2$, and $KO_2$, exactly coinciding with $M_2$. The $OP_2$ and $MSK_2$ constituents likely arise from different nonlinear aspects of the hydrodynamics (Parker, 1991), although both are generated in shallow water.

The $KO_2$ tide is important only when $M_2$ is anomalously small; it can potentially induce an unusual nodal modulation (relative to the standard $M_2$ modulation), but it has no bearing on seasonality. The few additional compound tides within the group noted by Simon (2013) are from interactions involving the diurnal $S_1$, which is normally small and will be ignored. Thus, when nonlinear shallow-water processes are acting sufficiently to produce a noticeable $MSK_2$ (or $OP_2$), the effect of this will be equivalent to semiannual modulations in $M_2$ amplitude and/or phase.

## 2.3 Climate-induced modulations

The astronomical sidelines in Table 1 sit at discrete known frequencies. Compound tides are similar, although the number of them can sharply increase in shallow water. In contrast, climate-driven modulations are broadband. Although an annual cycle typically dominates, which is the justification for $MA_2$, $MB_2$, one might expect higher harmonics in some cases and also possibly a general smearing of observed spectral lines across the group (e.g., Munk et al., 1965).

Müller et al. (2014), building on an earlier study by Foreman et al. (1995), discussed the case of Victoria (Canada). There was a clear annual modulation in both amplitude and phase of $M_2$, but with an apparent second harmonic. Multi-year tidal analysis of the Victoria data (not shown) does show energy at the $MSK_2$ (or $OP_2$) frequency. Whether this is due to the compound tide(s) or to a true second harmonic below $MA_2$ is not clear. During tidal analysis it is sometimes possible to separate two constituents of identical frequency by exploiting their different nodal modulations (Parker et al., 1999), but in this case the amplitudes appear too weak to allow it.

The question of frictional compound tides versus higher harmonics of $MA_2$ and $MB_2$ is only one instance of the more general, and difficult, problem of deciphering the causes of climate-induced modulations, even when manifested by $MA_2$ or $MB_2$ acting alone (e.g., Pugh and Vassie, 1994).





### 2.4 Nomenclature

The constituent names in Table 1 follow both historical and current international conventions—for the latter, see the table main-
120  tained by the Tide, Water Level and Current Working Group of the International Hydrographic Organization.[1] The exception
is $\beta_2$ which does not (yet) appear in the working group's table. Godin's 1988 book has the same usage as the IHO; his earlier
1972 book left the smaller lines unlabeled (Godin, 1972, 1988). Regarding $\beta_2$, however, it is commonly employed in the Earth
tide community (e.g., Hartmann and Wenzel, 1995; Calvo et al., 2016; Ducarme and Schueller, 2018), and $\beta_2$ falls into the
obvious pattern of using the first four letters of the Greek alphabet for the four astronomical constituents.

The two climate constituents apparently began as $MA_2$ and $Ma_2$ (Corkan, 1934). That lost favor, probably because a speaker
cannot distinguish the two and also early computers employed only upper-case letters. $MB_2$ has been used for decades, at least
in British work.[2] Amin (1976) still used $Ma_2$, but by 1983 he too had switched to $MB_2$.

    Not included in Table 1 is the pair, $H_1$, $H_2$, for the two annual sidelines, which is starting to appear fairly often in the
literature, probably because it is used in a popular open-source software package. The use of $H_1$ for a semidiurnal wave is
certainly an oddity, as the tide community has used an integer subscript to denote tidal species since at least the end of the
nineteenth century (Darwin, 1883; Harris, 1895; Cartwright, 1999). It is not clear where the symbol originated. An early
(possibly first) use was by Pugh and Vassie (1976). Before that, in a discussion of shallow-water tides, Godin (1972, Table
2.14) used the label "(Horn)" for both sidelines and he cited a discussion by Horn (1960), although Horn himself left the
lines unlabeled. Perhaps "(Horn)" has morphed into $H_1$, $H_2$, or the labels merely reflect the argument differences of $\pm h$. An
important point, however, is that both Godin and Horn, as well as Pugh and Vassie, were referring to the two climate lines—
none of their arguments involved $p'$—whereas the $H_1$, $H_2$ in present-day use are substituting for the gravitational tides $\alpha_2, \beta_2$,
as their arguments do involve $p'$. In any event, the use of a wrong subscript should be discouraged.

## 3   Three examples

In this section an example of $M_2$ seasonality arising from each of the three categories—gravitation, frictional interaction, and
climate processes—is presented. For each tide gauge analyzed, a tidal solution based on a single inversion of many years of
data was computed, sufficient to obtain reliable estimates of all constituents in Table 1. Based on an appropriate set of the
estimated constituents (different for each category), the implied modulation of $M_2$ over one year was computed by complex
demodulation. This is then compared with results of a second tidal inversion (or rather a set of inversions) in which estimates
of $M_2$ were obtained for every month of the multi-year time series, from which monthly means were then computed. (Standard
errors in these monthly means were estimated from the standard deviations for each month, scaled by $1/\sqrt{n}$ for $n$ years of
data.)

---

[1]https://iho.int

[2]David Pugh has a copy of a November 1977 memorandum from David Cartwright, then director of the Bidston Observatory, proposing use of the modified
symbols $MA_2$ and $MB_2$, in "deference to Corkan's original work." He suggested similar notation for other constituents affected by seasonal variability, such
as $NA_2$, $NB_2$; these are now included in the IHO tables, as are $MA_4$, $MB_4$.





**Table 2.** Amplitude, Greenwich phase lag, and dimensionless admittance for the $M_2$ group at Port Orford, Oregon

| Tide | $A$ (cm) | $G$ (°) | $|Z|$ |
|------|---------|---------|-------|
| $MSK_2$ | 0.03 | 52.2 | – |
| $\Gamma_2$ | 0.24 | 218.1 | $1.24 \pm 0.20$ |
| $\alpha_2$ | 0.27 | 198.4 | $1.24 \pm 0.18$ |
| $M_2$ | 74.70 | 216.5 | $1.18 \pm 0.00$ |
| $\beta_2$ | 0.19 | 205.2 | $0.98 \pm 0.20$ |
| $\delta_2$ | 0.08 | 199.4 | $1.12 \pm 0.40$ |

The goal is to confirm that seasonality of $M_2$, as delineated by monthly mean estimates of amplitude and phase, can—at least in these cases—be accurately reproduced by the modulations from a particular set of spectral lines. Which lines are in play differ depending on the category of causation.

## 3.1 Astronomical modulations

It is actually not easy to find good examples of seasonal variability stemming solely from the purely astronomical constituents of Table 1. Any potential case must display fairly constant admittances across the group of gravitational constituents. Yet at most tide gauges one sees perturbations in the admittance, or one sees significant amplitudes in the compound tides.

The time series at Port Orford, Oregon, is one of the better examples. Tidal constants estimated from 26 years of data[3] (1994–2021) are listed in Table 2. The magnitudes of tidal admittances $|Z|$ are all consistent within error limits, all phases are close, and the compound $MSK_2$ is very small. Combining the harmonic constants of the three astronomical constituents $\alpha_2, \beta_2, \delta_2$ implies a seasonal modulation of $M_2$ given by the solid lines of Fig. 2. These are in good agreement with the monthly mean estimates, aside perhaps for the January amplitude.

The theoretical seasonal modulations, based on Eqs. (3–4) are shown as the dashed lines. The solid and dashed lines agree well in phase, less well in amplitude, at first glance. But note that the amplitude vertical axis spans only 1 cm, so in fact the amplitude agreement is also quite good, with all data implying very little amplitude modulation. The small differences in amplitude curves occur because the estimated admittances in Table 2 are not identical, simply due to unavoidable estimation error.

The analysis at Port Orford confirms that the astronomically induced seasonal modulation of $M_2$ results in almost no amplitude modulation, and a phase modulation of about $0.7°$, with a minimum in April and maximum in September.

---

[3]Data at Port Orford are available from 1978 to present, but the data before 1994 yield tidal estimates too erratic to use. Data after 1993 appear of good quality.





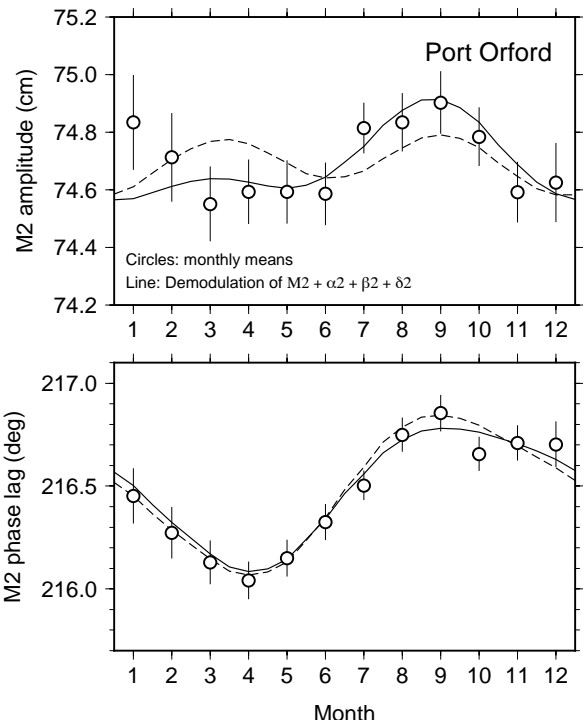

**Figure 2.** Seasonal variations in $M_2$ amplitude (top) and Greenwich phase lag (bottom) observed at Port Orford, Oregon. Circles with error bars are based on monthly mean $M_2$ estimates from 26 years of hourly data. The solid lines are the implied seasonal variations from the estimated side-constituents $\alpha_2, \beta_2, \delta_2$. The dashed lines are a theoretical seasonal modulation based on Eqs. (3–4). Note the amplitude axis spans only 1 cm.

### 3.2 Frictional/advective modulations

An example of modulations dominated by one or both of the compound tides in the $M_2$ group is Saint-Malo, France, whose spectrum was shown in Fig. 1. The two annual constituents ($\alpha_2, \beta_2$ or $MA_2$, $MB_2$) are smaller, as the spectrum reveals, but still too large to ignore. The total modulation, based on four constituents, is shown in Fig. 3. The amplitude clearly reveals the
presence of a semiannual modulation, with a slow rise during the beginning of the year, then a rapid decay between July and October. The phase is dominated by the semiannual effect, with an annual contribution responsible for the September phase lag exceeding the earlier peak in March.

### 3.3 Annual climate modulations

Tide gauges with annual variations in $M_2$, and thus with sidelines dominated by $MA_2$ and/or $MB_2$ are easy to find. The case
of Victoria was already noted (Müller et al., 2014), partly for having a second harmonic in its modulation. However, there are many tide gauges where only the annual modulation presents itself. A fair number can be found along the coast of Japan, even





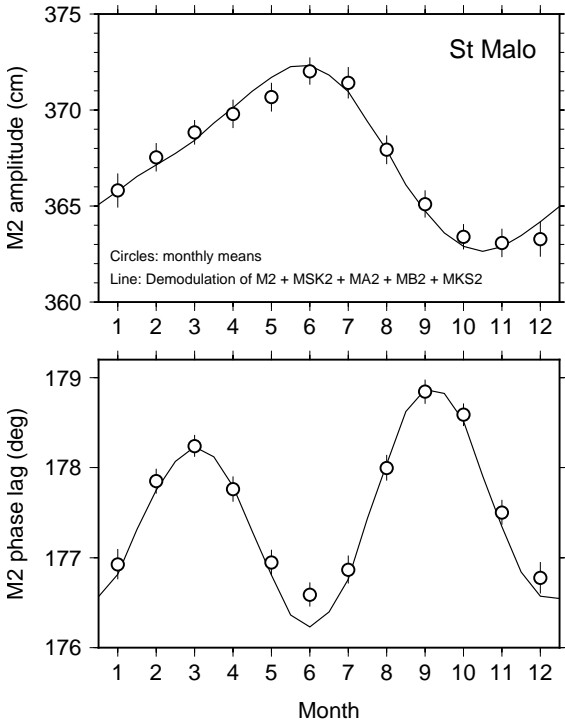

**Figure 3.** Similar to Fig. 2, but for Saint-Malo, France. The solid line is here based on demodulation of four tidal sidelines, the two annual constituents $MA_2$, $MB_2$ and the two compound constituents $MSK_2$, $MKS_2$. The latter dominate according to the spectrum shown in Fig. 1 and thus result in the clear semiannual modulation of the $M_2$ phase.

though $M_2$ itself is not especially large there. The example chosen here is in fact one of the largest modulations discovered anywhere: Chittagong, along the coast of Bangladesh. At that location the annual sea level term, Sa, is also anomalously large, presumably reflecting large discharge from the Ganges. Several other nearby tide gauges, with somewhat smaller $M_2$
modulations, were studied by Tazkia et al. (2017).

Harmonic analysis of eleven years of hourly data (2008–2018) at Chittagong reveals astonishing large amplitudes for $MA_2$ and $MB_2$ of 15.5 and 10.1 cm, respectively, with the $M_2$ constituent at 171.6 cm. The resulting seasonal modulation is shown in Fig. 4. The monthly mean amplitudes range from 146 cm in February to a high of 195 cm in August. In comparison, the phase modulation is not very large, about 4°. The monthly mean phases appear slightly erratic and fit the demodulated curve
only moderately well.

The large modulation in amplitude at this location closely mimics the large oscillation in annual sea level, which is minimum in February and maximum in July, with a mean range of 71 cm. Tazkia et al. (2017) developed a tide model for the region that explores this interdependence.

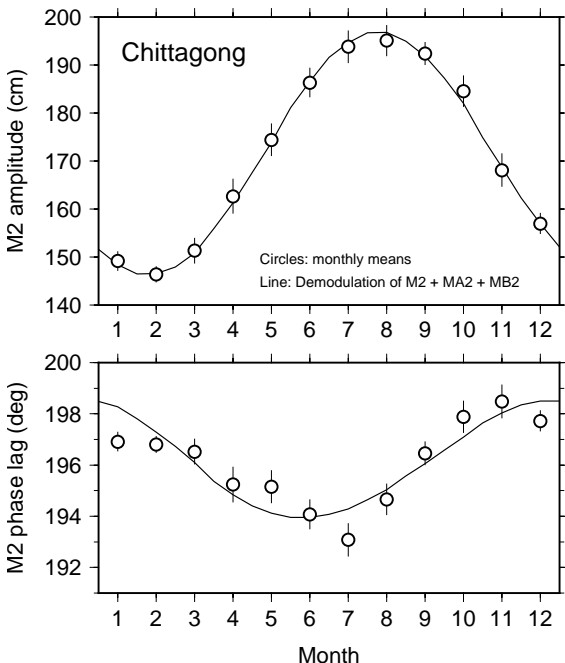

**Figure 4.** Similar to Fig. 2, but for Chittagong, along the coast of Bangladesh, east of the Ganges Delta. The unusually pronounced modulation of the $M_2$ amplitude at this location mimics the usually large annual oscillation of sea level, both having a minimum in February and maximum in July/August.

## 4 Conclusions

One indication of the variety of processes responsible for seasonal variability of the $M_2$ tide is the variety of different mechanisms generating spectral lines within the $M_2$ group: astronomical motions of the moon, frictional and other nonlinear interactions between tidal waves, and climate processes. The astronomical contribution is predictable given good long-term mean estimates of the $M_2$ constants; it is mostly a $\pm0.37°$ modulation in phase, precisely double the solar perturbation in the moon's longitude arising from the "annual equation" of lunar theory. On the other hand, when a tide gauge is found to be affected

by substantial seasonality in $M_2$, it usually arises from one or both of the constituents $MA_2$, $MB_2$. The variety of climate processes responsible for those two constituents—annual changes in stratification, sea level, ice cover, etc.—is where the real complication lies when attempting to understand seasonal variability.

*Data availability.* The Port Orford and Chittagong tide gauge data are available from the University of Hawaii Sea Level Center. The Saint-Malo tide gauge data are available from Le SHOM, the French national hydrographic service.





*Competing interests.* The author has no competing interests.

*Acknowledgements.* It is a pleasure to thank Philip Woodworth and David Pugh for discussions. This work was supported by the U.S. National Aeronautics and Space Administration through the Sentinel-6 and Sea Level Change projects.





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
