# Peer review of "Technical note: On seasonal variability of the M2 tide"

_EGUsphere, 2022_

## Author Response (AR1)

**Replies to Comments**

My responses are in **bold, red font.**

**Reviewer-1**

This manuscript is on the seasonal variability of the M2 tide, which has been widely recognized based on global observations. It provides a comprehensive and clear description of the origin of the seasonal variability of the M2 tide, which has not been found in literatures. In addition, the nomenclature used in the M2 tidal group is clarified, and H1 and H2 should be abandoned.

I believe that it will be a paper of high impact. The concept is fundamentally important to be written in the textbooks on the ocean tides. Thus, I recommand its publication in the current form.

**No response required.**

**Reviewer-2**

This manuscript on the seasonality of the M2 tide is an extremely relevant publication for the Ocean Sciences journal. Although, as the author states, the manuscript really summarises several well-known points and introduces evidence on these theories that date back to the early 1900s, the manuscript produces a comprehensive description that is of significant value to the tidal community. A general description of the tides, their sources and their relationships is in itself a valuable contribution and something often overlooked in the tidal community. I am very much a fan of Table 1 in providing very simple and important details. This manuscript will, therefore, be a valuable source of knowledge for the greater tidal community. Overall, the manuscript itself is a pleasure to read. Although, in my opinion, the manuscript is publishable as is, I do have a couple of comments which can hopefully clarify some points within the manuscript.

One point the author highlights is the length of the time series of data needed to separate the tidal constituents within the M2 tidal band. The explanations in Table 1 and Figure 1 demonstrate this nicely. I guess what is not clear, what is the implications when one does not have a long enough time series, on the estimation of the M2 and the overall tidal height prediction? Should one where possible directly estimate these sidelines and if not possible, what are the implications on the accuracy of tidal predictions? I realise the sidelines are usually fractions of the main M2, but the Chittagong application for example demonstrates significantly large modulations. This is of course more critical in tide gauges/bottom pressure sensors that have less than a year's worth of data or less frequent sampling patterns such as altimetry observations.

For the examples in the paper, I had purposely chosen tide gauges with many years of high-quality data. A long time series ensures that computed spectra (Figure 1) have adequate spectral resolution and that computations of monthly mean amplitudes and phases have relatively small error bars.  But there is no hard rule for the minimum amount of data needed before one can proceed.

Of course, it is implausible to study seasonal variability without data spanning most of a full year.  Multiple years are required if a computed spectrum is to have sufficient resolution to separate constituents with frequencies differing by 1 cpy. Spectral analysis is not mandatory, but it is certainly helpful to determine whether a spectrum contains isolated lines (as is the case for Figure 1) or instead is simply a wide cusp of energy surrounding M2.  One's interpretation of tidal variability would be sharply different in these two cases.

A simple Rayleigh criterion for separating constituents would also call for at least one year of data.  Yet that criterion is only a rough rule-of-thumb and depends on noise levels. Munk and Cartwright (1966) emphasized this by noting that it is possible to separate two nearby sine waves with only four perfect, noise-free observations, but with real-world noise something like a Rayleigh criterion is probably required. Of course, bottom pressure measurements of tides are usually much less noisy than surface height measurements, so there is flexibility in all this.

When reproducing the tide gauge evaluation in section 3, I found the same results as the author. However, in the period selected by the author, I noted a large temporal gap in the University of Hawaii dataset for Port Orford (shown below). I checked this with the PSMSL data (https://www.psmsl.org/data/obtaining/stations/1640.php) as well as GESLA-3 (flagged as -99 below) and UHSLC data. This is not a criticism of the results as these gaps in data are normal, but could this be an explanation for the differences seen in Figure 2? It could also be that the author has appropriate data from this tide gauge.

Yes, there are large gaps in the Port Orford time series before May 2002. These gaps could conceivably impact spectral analyses, but the gaps are followed by at least seventeen years without gaps, which are more than sufficient to support spectral calculations of good frequency resolution.  The calculations of monthly mean harmonic constants are unaffected, as each monthly mean is based on between 22 to 24 monthly estimates. Admittedly, the amplitudes in Figure 2 do not perfectly overlay. This likely stems as much from unavoidable estimation error (from inherently noisy measurements) as it does from any gaps.  Yet one should not belabor these small amplitude differences. In fact, the amplitudes differ by a span of only 3 mm, and so they are actually quite consistent.

**Addendum.**

Exchanges with Dr Pan are included in the earlier discussions.  I add the following:

The revised paper now acknowledges the possibility of seasonality in KO2 adding to seasonality in M2, in those rare cases where M2 is very small and KO2 unusually large. Finally, although not prompted by the reviews, I have inserted a sentence at the beginning of Section 3 which gives an additional technical detail on how monthly tides were estimated.